# A Focus on the Transformation Processes for the Valorization of Glycerol Derived from the Production Cycle of Biofuels

**Claudia Carlucci**

Department of Pharmacy—Drug Sciences, University of Bari "A. Moro", Via E. Orabona 4, 70125 Bari, Italy; claudia.carlucci@uniba.it; Tel.: +39-0805442251

**Abstract:** Glycerol is a valuable by-product in the biodiesel industries. However, the increase in biodiesel production resulted in an excess production of glycerol, with a limited market compared to its availability. Precisely because glycerol became a waste to be disposed of, the costs of biodiesel production have reduced. From an environmental point of view, identifying reactions that can convert glycerol into new products that can be reused in different applications has become a real necessity. According to the unique structural characteristics of glycerol, transformation processes can lead to different chemical functionalities through redox reactions, dehydration, esterification, and etherification, with the formation of products that can be applied both at the finest chemical level and to bulk chemistry.

**Keywords:** glycerol; biofuel; value-added products; continuous flow





## 1. Introduction

According to estimates by the World Energy Forum, 60% of the fossil fuels used in the field of energy production consumed in transport could run out in about a century, if alternative energy sources are not used, due to increasingly high industrialization and population growth [1].

One renewable and biodegradable source of energy is biodiesel, consisting of a mixture of methyl esters of fatty acids derived from several sources as edible and waste oil, algae, and fat. Biodiesel is a useful non-oil variant of fuel from an environmental point of view: engines have low emissions of unburnt hydrocarbons (<68%), carbon monoxide (44%), particulates (40%), sulfur oxide (100%), and polycyclic aromatic hydrocarbons (80–90%) [2].

Biodiesel can be obtained through a transesterification process starting from a mixture of triglycerides, such as vegetable or waste oils, by using alcohols. This reaction can be carried out in different conditions and with different kinds of catalysts (Figure 1) [3].

It is possible to identify four main types of catalysis: basic, acidic, and enzymatic catalysis, and the use of supercritical fluids. Transesterification is often carried out by basic catalysis, with the use of cheap and easily available strong bases such as NaOH, KOH, or NaOCH$_3$ [4].

The advantages of using this type of catalysis are reduced times (1–2 h), moderate temperatures (50–70 °C), easy recovery of glycerol, and the low cost of the catalyst.

The main disadvantages lie in the possibility of a saponification reaction which causes lowering yields, a difficult separation of glycerol, and a problematic washing of biodiesel, with excessive product loss due to the formation of emulsions. If the starting oil shows a free fatty acid content (FFA) greater than 3%, saponification can be checked. The best reaction conditions are achieved when the FFA content is less than 0.5%. This reaction is also influenced by the water content of the starting sample; indeed, prior to proceeding with the actual reaction, pretreatment can be performed. It is possible to perform a heating of the starting oil, steam injection, chromatographic column purification, or neutralization [5].

**Figure 1.** Transesterification reaction in biodiesel production.

Acid catalysis involves the use of catalysts such as $H_2SO_4$, HCl, and $H_3PO_4$. The advantage of this catalysis is a higher yield, due to the concomitant esterification reaction of the free fatty acids, but at the same time, resulted an increase in reaction times, temperatures and methanol/oil ratios compared to the basic catalysis [6].

A further disadvantage of both types of basic and acid catalysis lies in the non-reuse of the catalyst [7].

Another type of catalysis used recently involves a transesterification reaction by means of enzymes such as lipase, which hydrolyzes the triglycerides into glycerol and fatty acids through a lipolysis process. This reaction avoids the formation of by-products, produces high yields, and the glycerol is more easily recoverable, all at moderate temperatures and in short times, with recovery of the catalyst. The main disadvantage lies in the high cost of enzymes, which limits their use on a large scale [8].

It is also possible to use supercritical alcohols for transesterification, obtaining high yields in short times at the expense of drastic operating conditions, such as very high temperatures (239–385 °C) and pressures (200–400 bar). This technique is not given much consideration due to the high cost of the catalysts in relation to the yield of the reaction itself [9].

Besides the increase in the production of biodiesel on an industrial level, there is an overproduction of glycerol, a by-product of the transesterification reaction, which thus becomes waste material with a consequent increase in biodiesel production costs, due to disposal [10].

Glycerol (1,2,3-propantriol) is a colorless, odorless, viscous liquid with a sweet taste that is obtained from both natural and petrochemical raw materials [11].

Glycerol contains three hydrophilic alcoholic hydroxy groups, which are responsible for its solubility in water and its hygroscopic nature. In the aqueous phase, glycerol was stabilized by intramolecular hydrogen bonds and intermolecular solvation of the hydroxyl groups. Glycerol has a degree of purity ranging from 95 to 99%, while crude is between 70 and 80%. Glycerol is completely soluble in water and alcohols, slightly in many common solvents (ether, dioxane), and insoluble in hydrocarbons. It has a melting point of 18 °C and a boiling point at 290 °C under standard pressure conditions. It is widely used in personal care products, in the cosmetic field, in pharmaceutical formulations, and in foodstuffs; it is highly stable in storage conditions. Due to this property, glycerol is also added to adhesives and glues to slow down the drying process. It is also used as a solvent, sweetener, and preservative in food and drinks. We find it in medical and pharmaceutical preparations, such as a lubricant, humectant, emollient, and plasticizer in capsules and suppositories.

Glycerol has been also considered a valuable by-product in the industries that produce biodiesel, but the excess of its production compared to its availability has transformed it into a waste to be disposed of with the increase in the production costs of biodiesel [12].

## 2. Conversion of Glycerol under Batch Conditions

In order to produce eco-sustainable biodiesel, the search for reactions capable of converting glycerol into products has become a real need. Thanks to their unique structural characteristics, chemoselective transformations can lead to different chemical functionalizations, through redox processes, dehydration, esterification, and etherification, with products applied at the level of the finest chemistry up to "bulk chemistry" [13–17].

Figure 2 summarizes some of the main transformations of glycerol including that in lactic acid; it is a basic chemical compound particularly used in the synthesis of polymers and in the food and pharmaceutical industries.

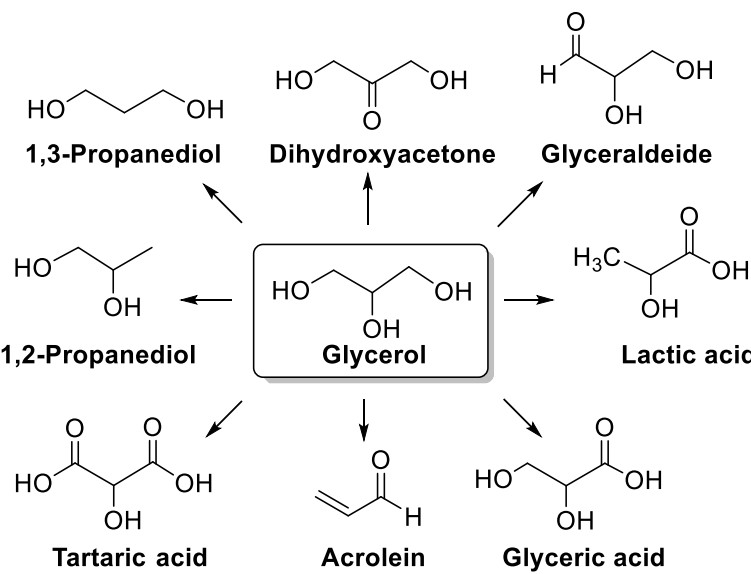

**Figure 2.** Conversion of glycerol into value-added products.

On the contrary, the use of crude glycerol deriving from the biodiesel industry is limited due to its composition which includes used catalysts, residual methanol, heavy metals, free fatty acids, etc. Its disposal is also difficult because, due to its methanol content, it is considered a hazardous waste.

Therefore, it is necessary to purify the raw glycerol, generally, through distinct steps such as neutralization to remove soaps and salts, vacuum evaporation to remove methanol and water, and refining to increase the purity of the glycerol. In the first step, the addition of acids converted the soaps into free fatty acids and, subsequently, following neutralization, a distinct third phase rich in salts and water is obtained. After separating the various phases obtained in the first step, alcohol and water were extracted through vacuum evaporation. At the end of this second step, the degree of purity of glycerol was about 85%. To obtain a quality product, further refining was carried out by means of various techniques such as steam distillation, ion exchange, or absorption of activated carbon [18].

Glycerol can be used in the production of monoglycerides; esterification with fatty acids, transesterification with methyl esters, or glycerolysis with vegetable oils were carried out. These reactions were catalyzed by bases, such as metal hydroxides or carbonates, or with sulfuric or phosphonic acids (Scheme 1) [19].

**Scheme 1.** Glycerol conversion into monoglycerides.

Polyglycerols were used in the production of non-ionic surfactants, involving allyl ethers, glycidol, or epichloroidine. The use of catalysts such as NaOH or KOH or carbonates, on the other hand, led to a non-selective polymerization of glycerol (Scheme 2).

**Scheme 2.** Polymerization of glycerol.

By means of etherification (Scheme 3), on the other hand, it was possible to obtain alkylglycerol ethers which were widely used in the pharmaceutical field for their anti-inflammatory [20], antibacterial [21], antifungal [22], anticancer [23], and immune system stimulant properties [24].

**Scheme 3.** Conversion of glycerol into ethers.

Costa explored acetins, mono-, di-, glycerol triacetates used as solvents, gelatinizers for explosives and perfume fixers, obtained by inserting a mixture of glycerol and acyl donor in the presence of appropriate enzymes (Scheme 4) [25].

**Scheme 4.** Conversion of glycerol into acetins.

Depending on the acyl donor, a greater selectivity towards mono-, di-, triacetin occurred. Acetic acid was more selective for monoacetin (46% after 24 h of reaction), vinyl acetate towards diacetin (98% after 1 h and 81% after 24 h, with production of triacetin as a by-product at 19% after 24 h), ethyl acetate towards monoacetin (42% after 1 h and 51% after 24 h, with production of diacetin at 16% after 24 h), and acetic anhydride, instead, turned out to be the donor that allowed conversion of 50% and 17% of monoacetin after 1 h and 24 h, respectively, 54% of diacetin, and 46% of triacetin after 24 h (Table 1).

The selective hydroxylation technique transformed the central hydroxy group into a tosyl group by removing the group obtained through catalytic hydrogenolysis. Alternatively, it was obtained through dehydroxylation, which took place after the acetalization of glycerol with benzaldehyde, tosylation of the hydroxyl group of acetylated glycerol and, finally, detosylation. This dehydroxylation took place with the support of transition metal catalysts such as Raney cobalt [17].

Other important methods of glycerol valorization were enzymatic conversion into value-added products and the chemical synthesis of epichlorohydrin through chemical processes [26,27].

**Table 1.** Acetins production under batch conditions.

| Acyl Donor [a] | Reaction Time | Conversion (%) [a] | Selectivity (%) [a] | | |
|---|---|---|---|---|---|
| | **(h)** | **(Yield %)** | **Mono** | **Di** | **Tri** |
| Acetic acid | 1 | 6 (2) | 6 | 0 | 0 |
| Vinyl acetate | 1 | 100 (92) | 0 | 98 | 2 |
| Ethyl acetate | 1 | 44 (38) | 42 | 2 | 0 |
| Acetic anhydride | 1 | 67 (58) | 50 | 17 | 0 |
| Acetic acid | 24 | 56 (50) | 46 | 10 | 0 |
| Vinyl acetate | 24 | 100 (92) | 0 | 81 | 19 |
| Ethyl acetate | 24 | 67 (60) | 51 | 16 | 0 |
| Acetic anhydride | 24 | 100 (94) | 0 | 54 | 46 |

[a] Reactions were conducted with 1:5 molar ratio between the glycerol and acyl donor at 60 °C with 20% (*w*/*w*) of Novozyme. Conversions were measured by GC–MS analysis as detailed in the experimental section and yields after product isolation.

Briggs reported the solventless process which converted glycerol to epichlorohydrin (GTE) in two chemical steps. In the first step, glycerol was hydrochlorinated with hydrogen chloride at elevated temperature and pressure, using a carboxylic acid catalyst, yielding a mixture of 1,3-dichlorohydrin, 1,3-dichloropropan-2-ol, 2,3-dichlorohydrin, and 2,3-dichloropropan-1-ol, while in the second step, the mixture of dichlorohydrins was converted to epichlorohydrin with a base (Scheme 5) [28].

**Scheme 5.** Conversion of glycerol to epichlorohydrin.

## 3. Biological Conversion of Glycerol Supported by Microorganisms

Many microorganisms were able to metabolize glycerol aerobically and anaerobically but none of them were used at industrial scale [29].

Luo [30] described how crude glycerol can be transformed into various products of biological interest such as 1,3-propanediol, *n*-butanol, 2,3-butanediol, docosahexaenoic acid (DHA), eicosapentaenoic acid (EPA), citric acid, lipids, and polyhydroxybutyrates (PHB) using different types of microorganisms (e.g., bacteria, fungi, microalgae) under anaerobic, microaerobic, or aerobic conditions (Table 2).

1,3-Propanediol could be obtained through anaerobic or microaerobic fermentation with different strains of *Clostridium* and *Klebsiella*. Wilkens showed that using pure glycerol, the strain *Clostridium butyricum* AKR102a reached 93.7 g/L of 1,3-propanediol, with a productivity of 3.3 g/L/h, while crude glycerol achieved 76.2 g/L and a productivity of 2.3 g/L/h [31].

Significantly high volumetric productivities have been achieved in batch and continuous cultures of the *Clostridium butyricum* strain carried out on glycerol derived from biodiesel production. Papanikolaou reported that the production of 35–48 g/L of 1,3-propanediol was the highest concentration achieved during the single-stage continuous cultures, with a maximum volumetric productivity obtained of 5.5 g/L/h [32].

**Table 2.** Biological conversion of crude glycerol into value-added products.

| Compound | Microorganism | Concentration (g/L) | Yield (g/g) | Productivity (g/L/h) |
|---|---|---|---|---|
| 1,3-Propanediol | *Clostridium butyricum* *Klebsiella pneumoniae* | 35–93.7 | 0.51–0.55 | 2.3–8.8 |
| *n*-Butanol | *Clostridium pasteurianum* | 6–12 | 0.19–0.44 | 0.03–0.15 |
| 2,3-Butanediol | *Klebsiella pneumoniae* *Klebsiella oxytoca* *Bacillus amyloliquefaciens* | 43–132 | 0.39–0.44 | 0.45–0.84 |
| Lactic acid | *Klebsiella pneumoniae* *Enterococcus faecalis* *Escherichia coli* | 50–142.1 | 0.82–0.93 | 0.77–2.96 |
| DHA | *Aurantiochytrium limacinum* *Schizocytrium limacinum* | 2–5 | - | 0.026-0.038 |
| EPA | *Pythium irregular* | 0.09 | - | $6.21 \times 10^{-4}$ |
| | *Mortierella ramanniana* *Mortierella isabelline* | 4–5.4 | 0.16–0.22 | - |
| Lipid | *Cryptococcus curvatus* *Chlorella protohecoides* | 23–24.6 | 0.31 | 0.06–0.13 |
| | *Yarrowia lipolytica* *Rhodosporidium toruloides* | 3.5–12.5 | 0.16 | 0.12 |
| PHB | *Zobellella denitrificans* *Cupriavidus necator* | 38.1–54.3 | 0.25 | 1.09–1.10 |

Menzel obtained, in a continuous fermentation of glycerol by *Klebsiella pneumoniae*, a final 1,3-propanediol concentration of 35.2–48.5 g/L, close to the maximum propanediol concentration found in batch and fed-batch cultures, while the volumetric productivity was about 2–3.5-fold higher in a range between 4.9 and 8.8 g/L/h [33].

By carrying out anaerobic fermentation by means of *Clostridium pasteurianum*, *n*-butanol was obtained. This conversion took place in 48 h with a productivity of 0.03–0.15 g/L/h, starting from a solution with 60 g/L of raw glycerol and 25 g/L of brown sugar bagasse [34–36].

The synthesis of 2,3-butanediol by means of various types of fermentation of crude glycerol with the support of bacteria such as *Klebsiella* or *Bacillus* with variable pH has been reported. It was underlined as, at pH 8 (0.47 g/L/h), a higher productivity was obtained compared to pH 7 or 5 [37–39].

Under microaerobic fed-batch fermentation, starting from pure glycerol, in the presence of *Klebsiella pneumoniae*, 142.1 g/L of optically pure D-lactate were accumulated with a yield of 0.82 g/g glycerol and a productivity of 2.96 g/L/h [40].

From fed-batch fermentation with 30 g/L glycerol and 10 g/L acetic acid, 55.3 g/L of L-lactic acid were obtained with a productivity of 0.77 g/L/h in the presence of *Enterococcus faecalis* (aerobic conditions) [41], while from 56 g/L of crude glycerol, 50 g/L of L-lactate were produced in the presence of an *Escherichia coli* strain (anaerobic) [42].

*Aurantiochytrium limacinum* and *Schizocytrium limacinum* are marine microorganisms that are used for their usefulness in the production of polyunsaturated fatty acids such as docosahexaenoic acid (0.026–0.038 g/L/h) [43–45], while *Pythium irregulare* leads to the synthesis of eicosapentaenoic acid ($6.21 \times 10^{-4}$ g/L/h) [46].

In comparison with glucose, the growth of various types of higher or lower fungi is more restricted on glycerol, which in many cases has been revealed as a not satisfactory substrate, potentially due to poor regulation of the enzymes involved in the first metabolic steps [47].

Glucose was a more relevant substrate for lipid accumulation than glycerol, with the exception of *Mortierella ramanniana* MUCL 9235, producing 4 g/L of lipids during growth on biodiesel-derived glycerol with a conversion yield of total lipid produced per unit of glycerol of 0.16–0.22 g/g. *Mortierella isabellina* ATHUM 2935 showed a high promising lipid production on both glucose and glycerol, producing 5.4 g/L of lipids, mostly triacylglycerols, with conversion yields per unit of glycerol consumed of 0.22 g/g [48].

A further way of recycling glycerol led to the production of lipids with the use of microalgae, yeasts, and fungi such as *Chlorella protothecoides* or *Cryptococcus curvatus*, showing how, in this case, fermentation in a fed-batch system, in which the volume was variable due to the possibility of adding more nutrients to the culture during fermentation, was more efficient compared to the batch mode (0.06–0.13 g/L/h) [49–51].

Papanikolau presented remarkable amounts of reserve lipid up to 3.5 g/L, produced by *Yarrowia lipolytica* grown on industrial glycerol, in highly aerated continuous cultures with a maximum volumetric productivity obtained of 0.12 g/L/h [52].

Diamantopoulou reported that *Rhodosporidium toruloides* DSM 4444 presented maximum lipid-accumulating capacities of 12.5 g/L and conversion yield on glycerol of 0.16 g/g [53].

The fermentation of glycerol, through a biological transformation, to produce polyhydroxyalkanoates (generally polyhydroxybutyrate), thermoplastic polymers used in the production of bioplastics (1.09–1.10 g/L/h), has been described [54,55].

A promising approach described the conversion of pure and crude glycerol into citric acid (CA) by *Yarrowia lipolytica* (Table 3).

**Table 3.** Biological conversion of glycerol into citric acid (CA) by *Yarrowia lipolytica*.

| Source | Microorganism | Concentration (g/L) | Yield (g/g) | Productivity (g/L/h) |
|---|---|---|---|---|
| Waste glycerol | *Y. lipolytica* A-101-1.22 | 112 | 0.6 | 0.71 |
| Waste glycerol | *Y. lipolytica* A-101-1.22 RB variant | 124.2 | 0.77 | 0.85 |
| Glycerol | *Y. lipolytica* NG40/UV5 | 87 | 0.64 | 0.906 |
| Waste glycerol | *Y. lipolytica* NG40/UV5 | 100 | 0.9 | 1.04 |
| Glycerol | *Y. lipolytica* LMBF Y-46 | 101.3 | 0.46 | - |
| Waste glycerol | *Y. lipolytica* LFMB 20 | 42 | 0.39 | - |
| Waste glycerol | *Y. lipolytica* ACA-DC 50109 | 62.5 | 0.56 | - |
| Glycerol | *Y. lipolytica* AWG7 | 154 | 0.74 | 1.05 |

Rymowicz studied the production of citric acid (112 g/L) by *Y. lipolytica* A-101-1.22, with a yield of 0.6 g/g and a productivity of 0.71 g/L/h, starting from waste glycerol derived from biodiesel industry. Active biosynthesis, using cell recycling prolongated up to 300 h, produced citric acid (96–107 g/L) with a yield of 0.64 g/g and a productivity of 1.42 g/L/h. The RB variant of 30% feed every 3 days showed the best results: 124.2 g/L citric acid with a yield of 0.77 g/g and a productivity of 0.85 g/L/h [56].

Morgunov showed that the mutant *Y. lipolytica* NG40/UV5 was able to synthesize CA in a medium with both pure glycerol and waste glycerol derived from biodiesel. *Y. lipolytica* NG40/UV5 produced 87 g/L of CA, while waste glycerol for *Y. lipolytica* NG40/UV5 cultivation increased CA production up to 100 g/L. Productivity was high and reached 0.906 and 1.04 g/L/h and the mass yield reached 0.64 and 0.9 g/g in the media with pure and waste glycerol, respectively [57].

Papanikolaou reported that *Y. lipolytica* LMBF Y-46 in aerated bioreactor experiments produced citric acid at 101.3 g/L, one of the highest in the literature for wild-type *Y. lipolytica* strains, in a successful fed-batch experiment with pure glycerol employed as the substrate [58], while *Y. lipolytica* LFMB 20 produced 42 g/L of citric acid and the conversion yield per unit of glycerol consumed was 0.39 g/g [48]. In another work, it was demonstrated that *Y. lipolytica* ACA-DC 50109 produced citric acid in conditions of low presence of nitrogen, with the highest quantity of 62.5 g/L and a yield of 0.56 g/g [59].

Rywińska reported that a mutant of *Y. lipolytica* Wratislavia AWG7 produced 154 g/L of citric acid in long-term repeated batch cultures, with a yield of 0.78 g/g and a productivity of 1.05 g/L/h, and the activity remained stable after 16 cycles in the repeated batch bioreactors [60].

He reported the path leading to the production of hydrogen, recognized as a potential source of energy and with the possibility of prolonged conservation, and highlighted how

it was possible to obtain 1.8 mmoles $H_2$/g of glycerol in fermented conditions in the dark from a culture of *Enterobacter aerogenes* and *Clostridium butyricum*, with the co-production of 1,3 propanediol, acetic acid, ethanol, and butyric acid. In the case of photofermentation, in a *Rhodopseudomonas palustris* culture, it was highlighted that 6.1 mmoles of $H_2$ were produced from 87% solution of crude glycerol [61].

Ethanol obtained as a by-product of the conversion of glycerol into hydrogen can be converted back into biodiesel synthesis processes. Jitrwung showed how crude glycerol can be converted into hydrogen and ethanol by *Enterobacter aerogenes* using a compound of $Na_2HPO_4$, $NH_4NO_3$, $MgSO_4$, $FeSO_4$, and $Na_2EDTA$ under semi-anaerobic conditions at pH 6.4, temperature of 37 °C, speed of 500 rpm mixing, and 0.44 mL/min average feeding speed (Table 4) [62].

**Table 4.** Comparison of media cost and hydrogen and ethanol yields for the conversion of crude glycerol in batch and continuous systems using various strains of *Enterobacter aerogenes*.

| Yields (mole/mole GL) | | Initial Glycerol | Reactor Type | SCAD/L of |
|---|---|---|---|---|
| $H_2$ | Ethanol | Concentration (g/L) | | Media |
| 0.96 | 0.90 | 15 | Batch–3.6 L | 0.91 |
| 0.86 | 0.75 | 15 | CSTR | 0.91 |

A fed-batch fermentation of *Klebsiella oxytoca* under anaerobic conditions allowed a yield of 0.20 g/g of ethanol from crude glycerol to be obtained, to which 1,3-propanediol was added as a by-product; an isolated strain of *Klebsiella pneumoniae* always provided 6.1 g/L of ethanol under anaerobic conditions; a significant yield of 27 g/L was obtained, instead, from an isolated strain of *Klebsiella cryocrescens* as also about 28.1 g/L were obtained by exploiting *Pachysolen tannophilus* and about 15 g/L, under strictly anaerobic conditions, from a culture of *Citrobacter freundii* [63].

Choi showed that *Kluyvera cryocrescens* S26 produced 27 g/L of ethanol from crude glycerol with a productivity of 0.61 g/L/h, in batch fermentation and under microaerobic conditions [64].

Oh reported a maximum production level of 21.5 g/L of ethanol produced from glycerol upon fermentation by the mutant strain GEM167 of *Klebsiella pneumoniae*. The production level was enhanced to 25.0 g/L upon overexpression of *Zymomonas mobilis* pdc and adhII genes, in the mutant strain. Both the mutant strains produced ethanol by crude glycerol, revealing a promising strategy of employing a waste byproduct as substrate at the industrial level [65].

The production of ethanol was also studied by Metsoviti, obtained from a strain of *Klebsiella oxytoca*, FMCC-197, on biodiesel-derived raw glycerol. The conversion of waste glycerol derived by biodiesel production produced 25.2 g/L of ethanol in fed-batch fermentations [66].

In another work, a new isolate *Citrobacter freundii* converted waste glycerol into ethanol in anaerobic batch bioreactor cultures. Ethanol was obtained with a concentration of 14.5 g/L, a conversion yield of 0.45 g/g, and a volumetric productivity of about 0.7 g/L/h [67].

## 4. Conversion of Glycerol Supported by Catalysts

Catalysts-based technologies have been deeply explored in the glycerol conversion process. Hydrocarbon-based conversion, including partial oxidation and a combustion reaction, has been promoted by metal, metal oxides, and metal sulfides. The development of a supported catalyst was aimed at reducing costs for large commercial applications. Among the inorganic supports available for preparing the support catalyst, there are silica, alumina, charcoal, zeolites, and other aluminosilicates, as well as more complex materials such as heteropolyacids [68].

Goetsch specified that the catalyst should preferably contain copper and zinc oxides. The synthesis of methanol preferably took place in a pressure range between 50 and 100 atm and the catalyst at a temperature between 176 and 246 °C. For a reaction with fewer by-products, a higher production of methanol, and a longer catalyst duration, it was also advisable that syngas entered the catalytic bed at a temperature between 200 and 250 °C [69].

Adhikari reported the synthesis of hydrogen by the steam reforming technique, an endothermic process that is performed at high temperatures where glycerol reacts with water according to the following reaction:

$$C_3H_8O_3 + 3H_2O \rightarrow 3CO_2 + 7H_2 \ (\Delta H^0{}_{298} = +123 \ kJ/mol)$$

This reaction was catalyzed by Ni/MgO, Ni/TiO$_2$, and Ni/CeO$_2$ at very high temperatures (650 °C) with the Ni/MgO catalyst providing the maximum yield (56%) of four moles of H$_2$ on the maximum possible seven stoichiometrically (Table 5) [70].

**Table 5.** Glycerin conversion and conversion into gaseous products over Ni-supported catalysts.

| Temperature (°C) | Ni/MgO | | Ni/CeO$_2$ | | Ni/Ti$_2$O | |
|---|---|---|---|---|---|---|
| | Glycerin Conversion | Conversion to Gases | Glycerin Conversion | Conversion to Gases | Glycerin Conversion | Conversion to Gases |
| 650 | 100 | 35 | 64 | 18 | 2 | 16 |
| 600 | 100 | 46 | 63 | 19 | 2 | 16 |
| 550 | 100 | 44 | 58 | 17 | 4 | 21 |

A further derivative, glycerol carbonate, was one of the substances that had excellent properties such as low toxicity, good biodegradability, and a high boiling point. It was used in gas separation membranes, polyurethane foams, surfactant components, and in the paint industry, produced through the reaction of glycerol with phosgene, transesterification with dialkylcarbonate or alkylencarbonate, carbonylation with urea, and reaction with carbon monoxide and oxygen in the presence of catalysts. The by-product obtained was ammonia in the gas phase, which was easily captured. Different catalysts were used: ZnSO$_4$, MgSO$_4$, La$_2$O$_3$, gold nanoparticles, Co$_3$O$_4$/ZnO, MgO, CaO, mixed metal oxides such as Al/MgOx, Al/LiOx, and also titanium catalysts with tungsten oxide (Scheme 6) [71].

**Scheme 6.** Synthesis of glycerol carbonate catalyzed by TiO$_2$.

Su analyzed the production of glycerol carbonate without the use of metals starting from glycerol, cyanopyridine, and carbon dioxide. The results obtained show that the best yields are obtained starting from 2-cyanopyridine compared to the 3- and 4-cyanopyridine isomers (Scheme 7) [72].

**Scheme 7.** Metal-free catalytic conversion of glycerol to glycerol carbonate.

Glycerol is also used in the production of acrylic acid, passing through the intermediates acrolein, propylene, allyl alcohol, lactic acid, 3-hydroxypropionic acid, and also acrylonitrile. Acrolein is widely used as an intermediate in the chemical and agricultural industry, in the production of DL-methionine, 1,2,6-hexantriol, quinoline, pentaerythritol, epoxy resins, and chemicals for water treatment, while crude acrolein is used in the acrylic acid synthesis chain. The dehydration of glycerol for the synthesis of acrolein begins with the protonation of the second hydroxyl group of the sugar with the resulting intermediate, which releases $H_3O^+$ to form 1,3-dihydroxypropene, followed by a keto-enolic arrangement to give 3-hydroxypropionaldehyde which, after further dehydrogenation, leads to the formation of acrolein, using heteropolyacids, zeolites, metal oxides, and phosphates as catalysts (Scheme 8).

**Scheme 8.** Conversion of glycerol to acrolein.

Sun described catalytic deactivation due to the formation of coal from the polycondensation of glycerol and acrolein [73]. Acrolein can subsequently be oxidized to acrylic acid. The use of catalysts based on molybdenum vanadate determined a transformation of over 90% of the acrolein used in the reaction. Another way to produce acrylic acid from glycerol is the exploitation of allyl alcohol as an intermediate. The first step is deoxydehydration, which can be performed with different techniques: the reaction is carried out in a liquid phase at a temperature between 235 and 240 °C and with formic acid as a catalyst and hydrogen donor, with a yield between 89 and 99%; another option uses a liquid phase between 140 and 170 °C with rhenium-based catalysts and an alcohol as a hydrogen donor. The second step consists of an oxidation which highlights how the use of catalysts based on molybdenum vanadate or gold allow the transition from allylic alcohol to acrylic acid. Sun also described the passage from glycerol to acrylic acid through a propylene intermediate: by means of a liquid phase reaction, using a Fe-Mo/C catalyst, carrying out a hydrogenolysis of propylene glycol at 300 °C, 80 bar of $H_2$ pressure after 6 h, in which the starting molecule was transformed first into 1,2-propoanediol and then into 1-propanol before being dehydrated in propylene (Scheme 9).

**Scheme 9.** Conversion of glycerol to acrylic acid through a propylene intermediate.

Once propylene was obtained, this was first oxidized to acrolein using a catalyst based on bismuth molybdate and subsequently, using a catalyst based on molybdenum vanadate, acrylic acid was obtained. Another route described by Sun used lactic acid as an intermediate: catalysts based on precious metals such as platinum, palladium, and gold were used which support the dehydrogenation of glycerol into glyceraldehyde or dihydroxyacetone that were then converted into lactic acid by Bronsted bases such as NaOH or Lewis acids such as $AlCl_3$ and $TiO_2$. After the formation of lactic acid, dehydration occurred in order to obtain acrylic acid: the most used catalysts were based on phosphates, hydroxyapatite, and zeolite. The passage from glycerol to acrylic acid through the acrylonitrile, however, passed from the acrolein: by carrying out an amino oxidation with the $VSbNbO/Al_2O_3$ catalyst, in the presence of $NH_3$ and $O_2$ at 400 °C, obtaining a selectivity towards the acrylonitrile of 58%, with a conversion of 83%, and acrolein as the main by-product. The subsequent hydrolysis from acrylonitrile to acrylic acid was carried out in $H_2SO_4$ with the formation of the acid and its related esters.

Mohamed reported a study on the conversion of glycerol to methanol, using combinations of Nickel and Copper with the HSZM-5 zeolite as a catalyst loaded with different

percentages of the two metals in basic catalysis. The results indicated that zeolite with copper leads to a complete conversion of methanol and a 67% yield by weight. Ion exchange resins have been shown to provide a better yield of glycerol ethers than niobic acid (HNbO$_3$) and β-zeolite [74].

Voutchkova-Kostal described the catalytic transformation of glycerol into lactic acid by using homogeneous catalysts such as Ir(I), Ir(III), and Ru(II) N-heterocyclic carbene (NHC) sulfonate-functionalized complexes, without cosolvents, in the presence of an aqueous base. The reaction was conducted under conventional and microwave heating. The most active catalyst reaches KOH (1 equiv) and a time of flight of 45,592 h$^{-1}$ by using microwave and 3477 h$^{-1}$ by using conventional heating, with a constant rate for at least 8 h to achieve quantitative conversion of glycerol in only 3 h (Scheme 10) [75].

**Scheme 10.** Catalytic transformation of glycerol into lactic acid.

Lingaiah studied the selective hydrogenolysis of glycerol to propylene glycol over titania-supported Ru catalysts prepared by conventional impregnation (IM) and deposition–precipitation (DP) methods [76].

The catalyst preparation method can influence conversion and selectivity during glycerol hydrogenolysis. Catalysts prepared by the DP method showed higher conversion than catalysts prepared by the IM method. Low Ru content is enough to achieve maximum conversion if the catalyst is prepared by the DP method. The catalyst with low Ru content exhibited maximum activity, which is related to the nature of the Ru species. The supported titania provided a platform for good dispersion of nano-size Ru particles, which are responsible for high activity. The presence of residual chlorine has a detrimental effect on glycerol hydrogenolysis for supported Ru catalysts. The catalyst is active even when crude glycerol and glycerol with alkali salts are used. The catalyst showed similar conversion and selectivity upon reuse without loss of any activity and selectivity with intact morphology of the catalyst. The conversion of glycerol and its selectivity to 1,2-propanediol (1,2-PD) also depend on the reaction temperature, hydrogen pressure, reaction time, and glycerol concentration (Table 6).

**Table 6.** Hydrogenolysis of glycerol.

| Catalyst | Conversion | Selectivity (%) | | | |
|---|---|---|---|---|---|
| | (%) | 1,2-PD | EG | Acetol | Others |
| 1 Ru/TiO$_2$ (DP) | 35 | 64 | 18 | 2 | 16 |
| 2 Ru/TiO$_2$ (DP) | 46 | 63 | 19 | 2 | 16 |
| 2 Ru/TiO$_2$ (DP) [a] | 44 | 63 | 19 | 2 | 16 |
| 2 Ru/TiO$_2$ (DP) [b] | 42 | 59 | 22 | 2 | 17 |
| 5 Ru/TiO$_2$ (DP) | 44 | 58 | 17 | 4 | 21 |
| 5 Ru/TiO$_2$ (IM) | 31 | 49 | 24 | 2 | 15 |
| 7 Ru/TiO$_2$ (DP) | 40 | 64 | 18 | 7 | 11 |

Reaction conditions: glycerol conc.: 20 wt.%, H$_2$ pressure: 60 bar, reaction time: 8 h, reaction temperature: 180 °C, catalyst wt.: 6%. [a] Crude glycerol. [b] Glycerol with 5% sodium sulphate.

Hutchings investigated base-free selective oxidation of glycerol using Au–Pd–Pt nanoparticles supported on titanium dioxide and their corresponding bimetallic catalysts. It was demonstrated that Pd–Pt/TiO$_2$ was the most active catalyst with high selectivity to glyceric, tartronic, and hydroxypyruvic acids and dihydroxyacetone (Table 7) [77].

**Table 7.** Reuse data for base-free glycerol oxidation (24 h, 60 °C, substrate/metal ratio of 2728:1).

| Run | Conversion at 24 h (%) | Selectivity (%) [a] | | | | | | | CMB |
|---|---|---|---|---|---|---|---|---|---|
| | | GA | TA | GLA | OA | DHA | HPA | FA | |
| Standard | 18.0 | 64.7 | 1.1 | 2.7 | 0.1 | 30.2 | 1.2 | - | 95.2 |
| Reuse 1 | 18.4 | 60.3 | 0.6 | 0.6 | 0.1 | 35.0 | 1.0 | 1.4 | 88.2 |
| Reuse 2 | 7.7 | 49.7 | 3.1 | 1.8 | 0.5 | 42.9 | 1.1 | 0.9 | 94.7 |

[a] GA glyceric acid, TA tartronic acid, GLA glycolic acid, OA oxalic acid, DHA dihydroxyacetone, HPA hydroxypyruvic acid, FA formic acid, CMB carbon mass balance.

Felpin described the use of graphene for the acetalization of glycerol with both aldehydes and ketones (Scheme 11). The authors showed unexpected and uncovered properties of graphene that surpass the activity of sulfated graphene (GR-SO$_3$H) for the acetalization of glycerol in neutral conditions (Table 8). Recycling studies revealed the robustness of graphene under the experimental conditions after six cycles [78].

**Scheme 11.** Acetalization of glycerol with carbonyl compounds.

Wang reported the functionalization of graphene oxide (GO) with heteropolyacids (HPAs) and the catalytic activity in the conversion of glycerol into lactic acid. The HPAs were surrounded by the lipid bilayer through electrostatic interactions with protonated amine groups; they resulted highly resistant to environmental changes and prevented leaching from the GO. Glycerol (1 M) was converted in 3.5 h into lactic acid, under mild conditions, at 60 °C and 10 bar O$_2$, resulting in a 90% yield and a 97% conversion, with recycling up to fifteen times (Scheme 12) [79].

**Table 8.** Acetalization of glycerol with carbonyl compounds.

| Carbonyl Compound [a] | Catalyst | Time (h) | Selectivity 5-Members/6-Members | Yield (%) |
|---|---|---|---|---|
| Benzaldehyde | GR | 2 | 63/37 | 94 |
| Benzaldehyde | GR-SO$_3$H | 2 | 33/67 | 49 |
| Anisaldehyde | GR | 2 | 63/37 | 99 |
| *Trans* Cinnamaldehyde | GR | 2 | 66/34 | 89 |
| Furfural | GR | 2 | 68/32 | 85 [b] |
| Acetone | GR | 2 | >99/<1 | 76 |
| Acetone | GR | 2 | >99/<1 | 85 [c] |
| Acetone | GR-SO$_3$H | 14 | >99/<1 | 59 [c] |
| Acetone | - | 14 | - | 0 [c] |
| Acetophenone | GR | 2 | >99/<1 | 25 |
| Acetophenone | GR | 2 | >99/<1 | 72 [c] |
| Cyclopentanone | GR | 2 | >99/<1 | 99 |
| Cyclohexanone | GR | 2 | >99/<1 | 99 |

[a] Reaction conditions: glycerol (1 mmol), carbonyl compound (10 mmol), and catalyst (25 mg) were stirred at 100–120 °C. [b] 12.5 mg of GR was used. [c] Reaction carried out at 120 °C.

**Scheme 12.** Graphene catalyzed conversion of glycerol into lactic acid.

Haider and Hutchings described the transesterification of plant-derived triglycerides with methanol to produce biodiesel. The by-product crude glycerol reacted with water over basic or redox oxide catalysts to produce methanol and other useful chemicals, in high yields, in a one-step low-pressure process. Molecules containing at least two hydroxyl groups can be converted into methanol (Scheme 13) [80].

**Scheme 13.** One-step low-pressure process production of methanol.

Ren and Chaudhari described a nanocatalyst with extraordinary performance for converting biomass-derived polyols to chemicals. The authors describe a new strategy for synthesizing copper-based nanocatalysts on a reduced graphene oxide support, yielding highly active Cu-graphene catalysts with a turnover frequency of 33–114 mol/g atom Cu/h for converting glycerol to value-added chemicals, such as lactic acid, ethylene glycol, and 1,2-propanediol. Trace amounts of palladium remarkably enhanced the overall stability of the Cu nanocatalysts and resulted in the generation of hydrogen from polyols (Scheme 14) [81].

**Scheme 14.** CuPd-graphene nanocatalysts conversion of biomass-derived polyols to chemicals.

## 5. Products Arising from Glycerol by Continuous Flow Procedures

The transformation of glycerol into various products that can be reused can be carried out both through batch and continuous flow procedures [82].

Flow chemistry is nowadays widely applied in the preparation of organic compounds, drugs, natural products, and sustainable materials. Microreactors and vaporization technologies have played an important role in both academic and industrial research in recent years by offering a viable alternative to batch processes. The use of continuous processes, with micro- or meso-reactors, allow access to reaction conditions not accessible with traditional systems. The microfluidic system involves an optimization of the reaction parameters such as mixing, flow rate, and residence time. Furthermore, pressure and temperature can easily be controlled, in parallel with other conditions such as solvent, stoichiometry, and work-up operations [83].

The use of continuous processes, inside micro- and meso-reactors, allows access to a wider profile of pressure and temperature conditions and to explore some of them that cannot be used in traditional batch systems. Within flow microreactors, pressure and temperature technologies can be easily controlled, and the reactions are faster compared to the corresponding ones in batch systems, resulting in temporal, spatial, and energy efficiency. The high surface/volume area ratio in the microreactors allows rapid heat transfers and high thermal control, decreasing the risk of reaching temperatures that can lead to a series of drawbacks from the point of view of safety and effectiveness of the reaction itself. The flow chemistry, however, presents some drawbacks including the difficulty of using poorly soluble substances for which high dilutions and a careful choice of the solvent are adopted. This measure allows the guaranteeing of greater solubility of reagents, intermediates, and products, in order to avoid clogging of the system, a disadvantage linked to the reduced dimensions of the channels used (a few millimeters and volumes between μL and mL).

Biodiesel can be produced in flow reactors and examples of transesterification carried out by means of a continuous process are reported in the literature [84].

In the first reactor (R1), the esterification of free fatty acids with methanol is carried out and subsequently, transesterification follows in the second reactor (R2) using a homogeneous catalyst such as sodium hydroxide or sodium methylate with conversion generally over 98%. At least two reactors in series with intermediate glycerol separation should be used; the reaction mixture is then subjected to phase separation in a mixture of esters and glycerol, in unit S1 by decantation or centrifugation. The phase containing the glycerol is treated with acid for the removal of soaps and recovery of FFA and the methanol is recovered by evaporation and recycling. The esters follow the path of separation from methanol in the S4 unit, followed by the neutralization of the catalyst, and finally, the washing and evaporation of the biodiesel (Figure 3).

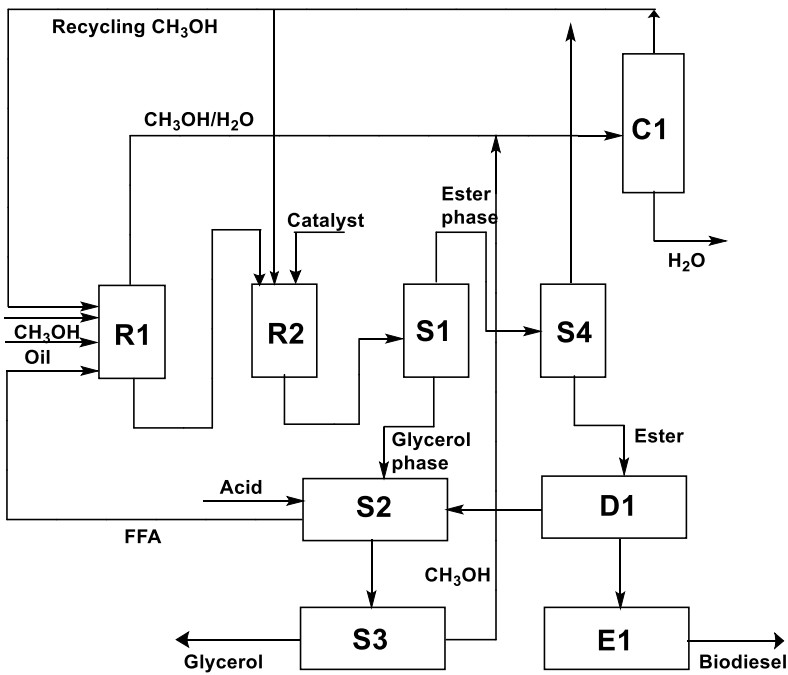

**Figure 3.** Biodiesel and glycerol produced in flow reactors.

Monbaliu described another transformation that represented a practical and economic alternative to the preparation of allyl alcohol, starting from glycerol, through a deoxy-dehydration (DODH). This reaction can be carried out under different conditions with different types of catalysis, including that with formic acid and metal-catalyzed reactions (Scheme 15) [85].

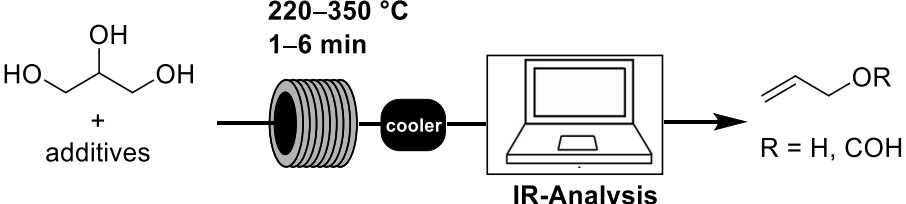

**Scheme 15.** Preparation of allyl alcohol from glycerol.

In order to be environmentally friendly, biodiesel production required an alternative production method of methanol that was not dependent on fossil fuels. From this perspective, one of the most interesting transformations was the use of glycerol to produce methanol, achievable once the glycerol was separated from biodiesel, through a process that consisted of different phases with percentage conversion for each step of about 25%. The unreacted product gases (CO, $H_2$, $CH_4$, $CO_2$) were separated and recycled in the reactor, while the methanol was purified and reused in the reactor where the transesterification occurred [86].

Chary exploited the hydrogenation of glycerol to form 1- and 2-biopropanol in a continuous process on a catalyst supported by titanium phosphatase. This catalyst shows excellent results with a constant selectivity of 97% towards propanol. The maximum conversion of glycerol is obtained at a temperature of 220 °C (Scheme 16). At high temperatures (280–400 °C) and at atmospheric pressure, it is possible to dehydrate the glycerol in acrolein by operating in a flow sequence. The first step of this dehydration of the glycerol in acrolein is carried out on a $WO_3/TiO_2$ catalyst with the injection of molecular oxygen to deactivate the catalyst. Then, the ammoxidation of atmospheric pressure is performed in fixed bed reactors with continuous flow connected to a stainless steel evaporator. The products are

then condensed into an acid solution to neutralize the unreacted ammonia and to avoid polymerization of the acrolein [87].

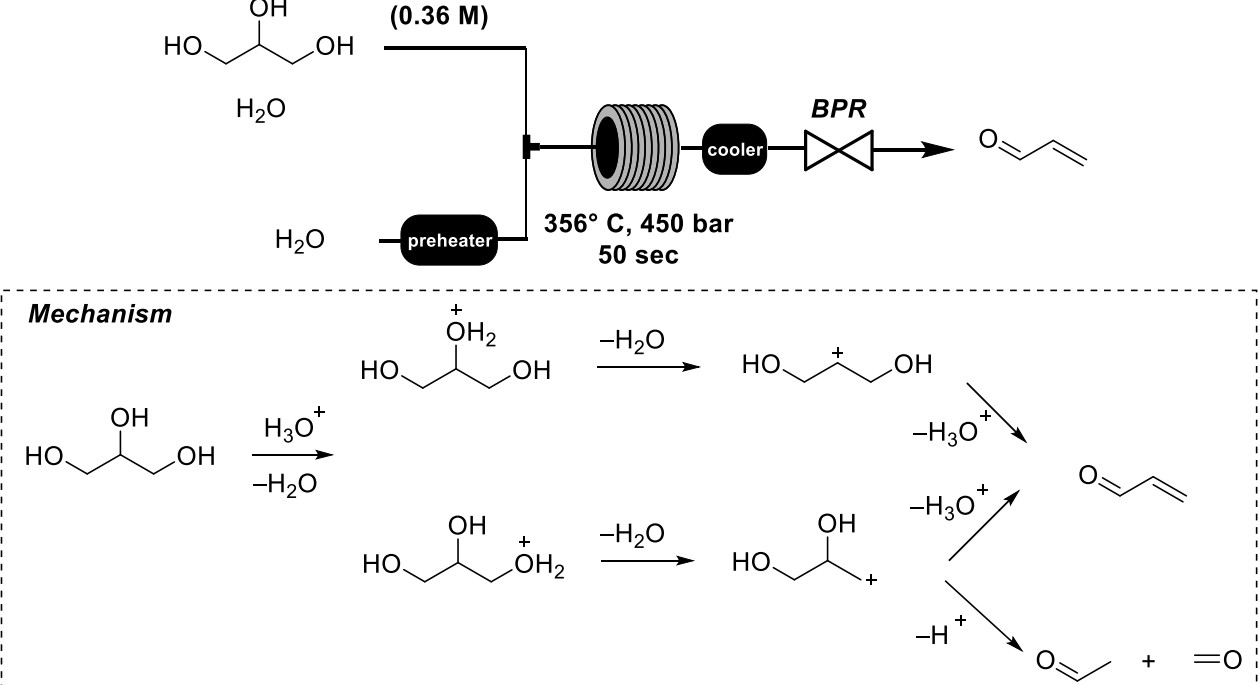

**Scheme 16.** Hydrogenation of glycerol to 1- and 2-biopropanol in a continuous process.

Costa used a 50 mL tube equipped with a stirring bar operating throughout the process to obtain the conversion of glycerol into mono-, di-, and triacetin [25]. An Asia Flow Reactor was equipped with an Omnifit column containing a lipase. The conditions of 60 °C and a flow rate of 0.4–3.0 mL/min were selected to verify the conversion obtained. The authors opted to use ethyl acetate and vinyl acetate as acyl donors and noted that ethyl acetate provides us with the same proportions as the batch conditions. With vinyl acetate, on the other hand, an increase in triacetin production has been noted.

The same flow study was carried out, then, without purifying the glycerol: the use of ethyl acetate led to a greater selectivity towards monoacetin. The use of vinyl acetate, on the other hand, led to greater selectivity for triacetin (84%) and, by increasing flow rate and residence time, diacetin is the most produced compound [88].

Some research described the synthesis of acrolein in the liquid phase, starting from glycerol with the use of continuous flow processes [89].

Kruse reported that it was possible to carry out the double dehydration of the glycerol using sub- and supercritical water as a reaction medium [90]. A supercritical solvent was obtained with temperatures and pressures higher than its critical point (for water 374 °C and 218 atm), while it was subcritical if the temperature and/or pressure did not exceed it. Scheme 17 shows the conversion of glycerol to acrolein with the use of subcritical water.

**Scheme 17.** Conversion of glycerol to acrolein with the use of subcritical water.

In Scheme 18, on the other hand, the conversion of glycerol into supercritical water was highlighted. In addition to acrolein, the formation of other products was also found, including acetaldehyde, formaldehyde, allyl alcohol, propane, and methanol.

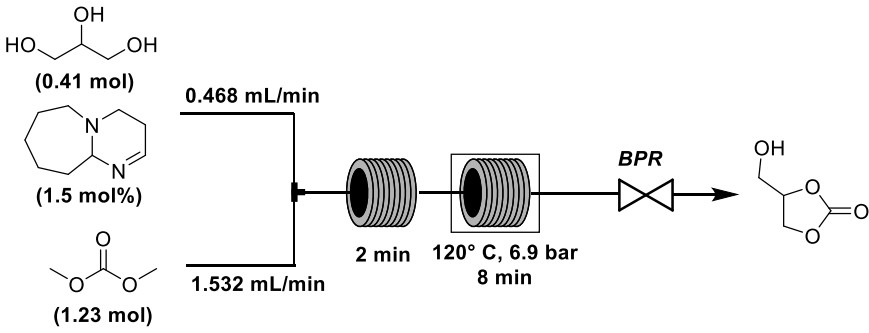

**Scheme 18.** Conversion of glycerol to acrolein with the use of supercritical water.

By means of oxidation in continuous flow, it was possible to obtain lactic acid or dihydroxyacetone, depending on the hydroxyl group affected by the reaction. Goto reported the oxidation of glycerol under hydrothermal electrolysis in the presence of NaOH as the base catalyst, demonstrating that, in addition to lactic acid and dihydroxyacetone, lower quantities of glycolaldehyde, formic, and acetic acid were also obtained in subcritical water in a continuous flow reactor equipped with titanium electrodes [91].

The main route of synthesis of glycerol carbonate occurred through the carbonation of glycerol with dimethyl carbonate which were, however, immiscible with each other. This problem can be overcome with the addition of 1,5-diazabicicloundec-7-ene (DBU) or by using methanol as solvent [92]. A possible mechanism for carbonatation of glycerol in the presence of DBU as a base catalyst is represented in Scheme 19.

**Scheme 19.** Carbonatation of glycerol.

The oxidation of glycerol to high-value carboxylic acids in a liquid-phase flow reactor was reported by Mimura who developed catalysts based on Au nanoparticles dispersed on alumina ($Al_2O_3$) support. The catalysts were prepared through a deposition–precipitation method, using an alumina support with high temperature stability and a particle size of $50-200$ µm. The glycerol was converted up to 70% into glyceric and tartronic acid and the catalytic activity was up to 1770 min. To increase the activity, the catalyst was pretreated with fructose (Scheme 20) [93].

**Scheme 20.** Oxidation of glycerol on an alumina ($Al_2O_3$) support-Au nanoparticles.

Selva reported the acetalization of six different types of glycerol under continuous flow conditions, at 10 bar and 25 °C, using Amberlyst 36 and aluminum fluoride three hydrate ($AlF_3 \cdot 3H_2O$) as catalysts. Conversion (85%–97%) and selectivity (99%) were very high by using either pure or wet glycerol as the reagent. On the contrary, the continuous flow acetalization of crude-like glycerol was irreversibly deactivated in a few hours by the presence of low amounts of inorganic impurities of raw glycerol, while the acetalization of crude glycerol, at 10 bar and 100 °C with acetone and 2-butanone, catalyzed by aluminum fluoride, occurred with a productivity up to 78% and 5.6 h$^{-1}$ (Scheme 21) [94].

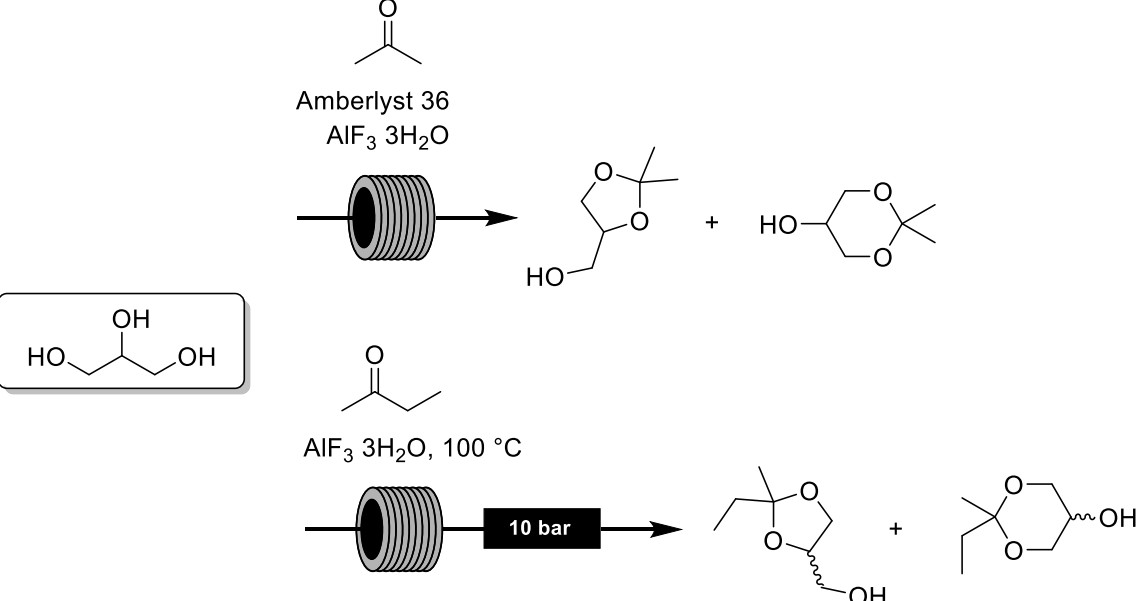

**Scheme 21.** Acetalization of glycerol under continuous flow conditions.

## 6. Conclusions

Public attention to environmental protection, energy consumption, and emissions of polluting elements is increasing. The constant increase in the costs of raw materials obtained from oil and the environmental impact of production processes have given

considerable impetus to the search for new materials from renewable sources and to the proposal of solutions that allow energy consumption to be decreased and reduce both the use of hazardous substances and the production of waste, promoting a sustainable development model. The transformation processes that allowed the recovery and reuse of glycerol, which is currently a by-product of the biodiesel production process, were explored. The glycerol could be transformed into high added-value chemicals or as an alternative source of molecular hydrogen.

A further detailed aspect concerned the transformation of glycerol by using microtechnologies. Flow chemistry, in fact, represents a useful technology to reduce the environmental impact of chemical processes, acting as a valid alternative to traditional processes and offering, in many cases, more sustainable synthetic pathways. Microtechnologies allow the achievement of reproducible reactions which are safer, with reduced costs, and with a significant reduction in by-products.

**Funding:** This research received no external funding.

**Acknowledgments:** This work was supported by the Intervento cofinanziato dal Fondo di Sviluppo e Coesione 2007–2013—APQ Ricerca Regione Puglia "Programma regionale a sostegno della specializzazione intelligente e della sostenibilità sociale ed ambientale—FutureInResearch".

**Conflicts of Interest:** The author declares no conflict of interest.

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
