# Peer review of "A Focus on the Transformation Processes for the Valorization of Glycerol Derived from the Production Cycle of Biofuels"

_catalysts, doi:10.3390/catal11020280_

Round 1

Author Response

Response to Reviewer 1 Comments

Title: A Focus on the Transformation Processes for the Valorization of Glycerol Derived from the Production Cycle of Biofuels I have appreciated the re-submitted manuscript 1121739, which is the revised version of the paper 1080080. Most of the points raised in the initial version of the manuscript have been sufficiently and satisfactorily addressed. Some points of relatively minor importance need to be revised:

1) Several figures and schemes (i.e. Fig. 1, Scheme 1, 5, 6, 7, etc) need legends.

2) I would greatly appreciate to see the reactions governing the chemical synthesis of epichlorohydrin from glycerol.

3) Tables 2, 3 and 4 request revisions. All names of microorganisms must be put in italics. Also, the names of the species (i.e. lipolytica, coli, etc) should not begin with capital letter.

4) In all text, names of microorganisms must be put in italics.

Decision: Minor revision based on the referee’s comments is requested for the current submission

Response: I thank the Referee for his appreciation regarding the modified version of the manuscript. I have added the legend to all the schemes and figures. I have described the synthesis of epichlorydrine in the new scheme 5. The names of the microrganisms within the tables and text were modified, accordingly.

Reviewer 2 Report

Interesting topic, dealing with an emergent issue.

Just to know how much of the cinsumed fuels worldwide is the share of biodiesel (production/consumption). This inofrmation would be interesting to know just to frame the amount of glycerol as byproduct of biodiesel.

On the catalytic conversion of glycerol, enzymatic catalysts were not considered. Maybe a reference to this method could be given.

Author Response

Response to Reviewer 2 Comments

Interesting topic, dealing with an emergent issue.

Just to know how much of the consumed fuels worldwide is the share of biodiesel (production/consumption). This information would be interesting to know just to frame the amount of glycerol as byproduct of biodiesel.

On the catalytic conversion of glycerol, enzymatic catalysts were not considered. Maybe a reference to this method could be given.

Response: I thank the Referee for his suggestion. According to Renewables 2019 Global Status Report, global production of biodiesel was estimated to be 41.3 billion litres per year (34.3 from FAME and 7 from HVO). On the catalytic conversion of glycerol, the required reference [26] was already included in the bibliography, I have integrated the enzymatic catalysis within the text.

Reviewer 3 Report

The manuscript A Focus on the Transformation Processes for the Valorization of Glycerol Derived from the Production Cycle of Biofuels is very well written, clear and concise and my opinion is that it can be published in its current form without further corrections.

Author Response

Response to Reviewer 3 Comments

The manuscript A Focus on the Transformation Processes for the Valorization of Glycerol Derived from the Production Cycle of Biofuels is very well written, clear and concise and my opinion is that it can be published in its current form without further corrections.

Response: I thank the Referee for his kind appreciation and positive feedback.

This manuscript is a resubmission of an earlier submission. The following is a list of the peer review reports and author responses from that submission.

Round 1

Reviewer 2 Report

The manuscript entitled " A Focus on the Transformation Processes for the Valorization of Glycerol Derived from the Production Cycle of Biofuels" describes the synthesis of value-added chemicals from glycerol. This work is lack of novelty and several (and several) reviews ( some examples : Energy Environ. Sci., 2018, 11, 1012-1029; Catalysts 202010(11), 1279; https://doi.org/10.3390/catal10111279; Chem. Rev. 2010, 110, 3, 1807) have been already published about this topic and for this reason I recommend to reject this work.